# Four Cycles of Docetaxel-Cyclophosphamide versus Anthracycline-Taxane as Adjuvant Chemotherapy for HER2-Negative, Axillary Lymph Node Negative Breast Cancer: A Real-World Comparison of Alberta Patients Treated 2008–2012

**Malek Hannouf [1], Atul Batra [2,3] and Sasha Lupichuk [2,\***

[1] Cumming School of Medicine, University of Calgary, Calgary, AB T2N 4N2, Canada; malek.hannouf@ucalgary.ca
[2] Department of Medical Oncology, Tom Baker Cancer Center, 1331 29 ST NW, Calgary, AB T2N 4N2, Canada; atul.batra@ahs.ca
[3] Department of Medical Oncology, All India Institute of Medical Sciences, New Delhi 110029, India
[\*] Correspondence: sasha.lupichuk@ahs.ca; Tel.: +1-403-521-3688

**Abstract:** Uncertainty exists around the need to include an anthracycline if taxane-based adjuvant chemotherapy is being used for human epidermal growth factor receptor-2 (HER2) negative and axillary lymph node negative (LNN) breast cancer. We identified all patients who were diagnosed with HER2-negative, LNN breast cancer treated with docetaxel-cyclophosphamide for four cycles (DC4) or an anthracycline-taxane (AT) regimen following surgical resection in Alberta from 2008 through 2012. We used propensity score methods to match each patient treated with AT to up to four patients treated with DC4 on potentially confounding clinicopathologic and treatment variables. We compared the 10-year invasive disease free survival (iDFS), breast cancer specific-survival (BCSS) and overall survival (OS) and assessed the effect of the type of adjuvant chemotherapy on these outcomes using Cox regression. Of the 726 eligible patients, 657 (90.5%) were treated with DC4 and 69 (9.5%) were treated with AT. Matching created a group of 202 women treated with DC4 and eliminated differences in clinicopathologic and treatment factors. There was no statistically significant difference for the treatment effects of matched DC4 patients compared to the AT patients on iDFS (75.7% vs. 76.8%, *p* = 0.75; hazard ratio (HR) = 1.05, 95% CI = 0.65 to 1.8), BCSS (88.1% vs. 87%, *p* = 0.8; HR = 0.91, 95% CI = 0.42 to 1.9), or OS (87.1% vs. 86.9%, *p*= 0.96; HR = 0.98, 95% CI = 0.46 to 2.1). Four cycles of DC as compared with an AT regimen yielded similar 10-year iDFS, BCSS and OS amongst patients with HER2-negative, LNN breast cancer.

**Keywords:** breast cancer; lymph node negative; HER2-negative; adjuvant chemotherapy; anthracycline; taxane; invasive disease free survival; breast cancer specific-survival; overall survival; propensity-score matching





## 1. Introduction

Adjuvant anthracycline-taxane (AT) combinations for human epidermal growth factor receptor-2 (HER2) negative breast cancer have been widely studied and adopted due to improved survival outcomes in comparison to historical anthracycline-only regimens [1–4]. However, anthracyclines are associated with small increases in the absolute risks for late, irreversible cardiotoxicity, myelodysplastic syndromes and leukemias [5]. Docetaxel plus cyclophosphamide for four cycles (DC4) has been compared with the anthracycline-only regimen and doxorubicin plus cyclophosphamide for four cycles (AC4), and demonstrated superior disease-free survival (DFS) and overall survival (OS) [6]. Subsequent trials have attempted to address the effectiveness of six cycles of DC (DC6) compared to the more widely used AT regimens [7–9]. In a recent pooled analysis of these studies, AT regimens

were not found to be superior to DC in terms of DFS or OS, while the non-inferiority of DC against AT for these outcomes continued to be ambiguous [10].

In Alberta, DC4 has been used as a standard adjuvant chemotherapy regimen for HER2-negative, lymph node negative (LNN) breast cancer. However, AT options have been available and the choice of regimen has been at the discretion of the treating medical oncologist in discussion with the patient. The objective of this study was to compare the 10-year survival outcomes of patients diagnosed with HER2-negative, LNN breast cancer who were treated with DC4 as compared to those treated with an AT regimen.

## 2. Materials and Methods

### 2.1. Data Sources and Identification of Study Population

As previously described [11], patients were retrieved from the Alberta Health Services (AHS) Cancer Control Breast Data Mart (BDM) and clinical variables were retrieved both from the BDM and through review of the electronic medical record. We included patients diagnosed with HER2-negative, LNN breast cancer diagnosed 1 January 2008 through 31 December 2012, who were prescribed DC4 or an AT regimen. For patients with hormone receptor (HR) positive breast cancer, the tumour had to be pT1c and grade III, or pT2-pT3 with any grade.

Ethics were institutionally approved under the Alberta Research Ethics Community Consensus Initiative [12].

### 2.2. Outcomes

The end-points for this study included 10-year invasive disease free survival (iDFS), breast cancer specific survival (BCSS) and overall survival (OS).

### 2.3. Statistical Analysis

Continuous data are reported as mean ± standard deviation, and categorical data as numbers and percentages. Categorical data were compared using the chi-square test. Quantitative variables were compared using the t-test. All statistical tests were two-sided and results were considered significant at the 5% critical level. Statistical analysis was performed using SAS, version 9.3 (Cary, NC, USA).

We performed a matched analysis within our cohort study. We used logistic regression to create a propensity score [13] for having an AT regimen, using the following potential confounders: age, year of diagnosis, body mass index, Charlson co-morbidity index (CCI) score, histology, stage, HR status, radiotherapy, type of breast cancer surgery, and time to adjuvant chemotherapy. We used the propensity score to match each patient who had an AT regimen with up to four patients who had DC4 on the estimated propensity score. To avoid a poor quality match, we only considered observations that were within a ±0.01 of the AT regimen unit's propensity score for matching and chose the closest match without replacement (i.e., caliper matching without replacement) [13]. When no matches were found, that case would be dropped.

Kaplan–Meier methods were used to determine iDFS, BCSS, and OS. Where appropriate, the log rank test was used to describe differences between survival curves. Cox proportional hazard modeling was used to calculate hazard ratios (HRs) with associated 95% CIs to assess the differences between the matched DC4 and AT patient groups with respect to 10-year iDFS, BCSS, and OS.

In separate analyses, we used Kaplan–Meier survival curves and Cox regression to compare the 10-year iDFS, BCSS, and OS in all patients with DC4 to the case group of patients with AT and generate HRs. We conducted standard adjusted analyses by including all potential confounders mentioned earlier as covariates in a Cox proportional hazard model. We also used the propensity score to adjust for differences in baseline characteristics between the two patient groups using two methods. First, we used the propensity score as a covariate in a Cox proportional hazard model and generated adjusted HRs [13]. Second, we

used a weighted Cox proportional hazards model and generated adjusted HRs, where the weight assigned for each patient was based on the stabilized inverse propensity score [14].

In all above analyses, we examined whether the association of adjuvant chemotherapy (DC4 vs. AT) with 10-year iDFS, BCSS, and OS is different by tumour characteristics (grade, stage and hormone receptor status) by testing for interactions between these tumour characteristics and the type of adjuvant chemotherapy (DC4 vs. AT).

## 3. Results

We identified 726 patients who were diagnosed with LNN, HER2-negative breast cancer during the period from 1 January 2008 to 31 December 2012 and met our study inclusion criteria. Of those, 69 were treated with an AT regimen (9.5%) and 657 (90.5%) were treated with DC4 (Table 1). The most commonly prescribed AT regimen (91.3%) was three cycles of fluorouracil, epirubicin (100 mg/m$^2$) and cyclophosphamide followed by three cycles of docetaxel (FEC-D). Prior to matching, patients treated with an AT regimen were more likely to be younger, have a lower Charlson co-morbidity score and mastectomy for stage II, grade 3 and hormone receptor-negative breast cancer (Table 1). Using 1:4 matching on the estimated propensity score, we matched the AT group of 69 patients with a DC4 group of 202 patients. No AT cases were dropped due to poor match quality. As a result of matching, we eliminated the previously described differences between the two groups (Table 1).

**Table 1.** Baseline patient, tumour, and treatment characteristics of patients in the anthracycline-taxane (AT), overall docetaxel-cyclophosphamide for four cycles (DC4) and matched DC4 groups.

| Characteristic | AT (n = 69) | DC4 (n = 657) | *p*-Value | Matched DC4 (n = 202) | *p*-Value |
|---|---|---|---|---|---|
| **Age (years)** | | | | | |
| Median | 46 | 53 | | 47 | |
| Mean (SD; range) | 45.8 (9.7; 25–69) | 52.6 (9.8; 23–78) | <0.0001 | 46 (9.4; 23–69) | 0.8 |
| **Body Mass Index (kg/m$^2$)** | | | | | |
| Median | 25.3 | 27.5 | | 26 | |
| Mean (SD; range) | 27.1 (6.7;17.5–62) | 28.5 (6.6;15.6–64) | 0.1 | 27.1 (5.8;17.3– 49.4) | 0.9 |
| **Year of diagnosis [n (%)]** | | | | | |
| 2007–2008 | 11 (15.9%) | 78 (11.9%) | 0.24 | 26 (13%) | 0.8 |
| 2009–2010 | 22 (31.9%) | 275 (41.9%) | | 68 (33.5%) | |
| 2011–2012 | 36 (52.2%) | 304 (46.3%) | | 108 (53%) | |
| **Charlson co-morbidity score [n (%)]** | | | | | |
| Mean (SD, range) | 0.04 (0.26; 0–2) | 0.33 (0.67; 0–4) | <0.0001 | 0.04 (0.27; 0–2) | 0.9 |
| Score > 0–no. of patients (%) | 2 (2.9%) | 151(23%) | 0.0005 | 7 (3.4%) | 0.8 |
| 0 | 67 (97.1%) | 506 (77%) | | 195 (96.6%) | |
| 1 | 1 (1.45%) | 97 (14.8%) | | 4 (2%) | |
| 2 | 1 (1.45%) | 44 (6.7%) | | 3 (1.4%) | |
| 3 | 0 | 9 (1.4%) | | | |
| 4 | 0 | 1 (0.15%) | | | |
| **Histology [n (%)]** | | | | | |
| Ductal | 58 (84%) | 479 (72.9%) | 0.12 | 165 (82%) | 0.8 |
| Mixed Ductal-Lobular | 5 (7.25%) | 96 (14.6%) | | 20 (9.9%) | |
| Others | 6 (8.7%) | 82 (12.5%) | | 17 (8.4%) | |
| **Stage [n (%)]** | | | | | |
| Stage I | 12 (17.4%) | 253 (38.5%) | 0.0005 | 37 (18.3%) | 0.9 |
| Stage II | 57 (82.6%) | 404 (61.5%) | | 165 (81.6%) | |
| **Grade [n (%)]** | | | | | |
| Well differentiated | 0 | 26 (4%) | <0.007 | 0 | 0.9 |
| Moderately differentiated | 6 (8.7%) | 139 (21.2%) | | 20 (9.9%) | |
| Poorly differentiated | 63 (91.3%) | 492 (74.9%) | | 182 (90.1%) | |
| **Hormone receptor status [n (%)]** | | | | | |
| Positive | 33 (47.8%) | 468 (71.2%) | <0.0001 | 95 (47%) | 0.9 |
| Negative | 36 (52.2%) | 189 (28.8%) | | 107 (52.9%) | |
| **Definitive breast surgery [n (%)]** | | | | | |
| Breast-conserving surgery | 30 (43.5%) | 390 (59.4%) | <0.01 | 90 (44.5%) | 0.9 |
| Mastectomy | 39 (56.5%) | 267 (40.6%) | | 112 (55.5%) | |

**Table 1.** *Cont.*

| Characteristic | AT (n = 69) | DC4 (n = 657) | *p*-Value | Matched DC4 (n = 202) | *p*-Value |
|---|---|---|---|---|---|
| **Radiotherapy [n (%)]** | 36 (52.2%) | 397 (60.4%) | 0.18 | 103 (51%) | 0.9 |
| **Time interval to first cycle of chemotherapy (months)** | | | | | |
| Median | 2.73 | 2.8 | | 2.71 | |
| Mean (SD; range) | 2.86 (0.77; 1.2–4.8) | 3.25 (1.38; 0–15) | 0.5 | 2.82 (0.8; 0–5.9) | 0.7 |
| No. of patients (%) | | | | | |
| ≥0 months–<3 months | 39 (56%) | 404 (61.5%) | 0.42 | 122 (60%) | 0.6 |
| ≥3 months–<6 months | 30 (43.4%) | 245 (37.2%) | | 80 (40%) | |
| ≥6 months | 0 | 8 (1.2%) | | 0 | |
| **Type of adjuvant chemotherapy [n (%)]** | | | | | |
| DC×4 cycles | | 657 (100%) | | 202 (100%) | |
| FEC × 3 cycles → D × 3 cycles | 63 (91.3%) | | | | |
| AC × 4 cycles → D × 4 cycles | 3 (4.35%) | | | | |
| DAC × 6 cycles | 3 (4.35%) | | | | |

Abbreviations: A = doxorubicin (Adriamycin); C = cyclophosphamide; D = docetaxel; E = epirubicin; F = 5−fluoruracil.

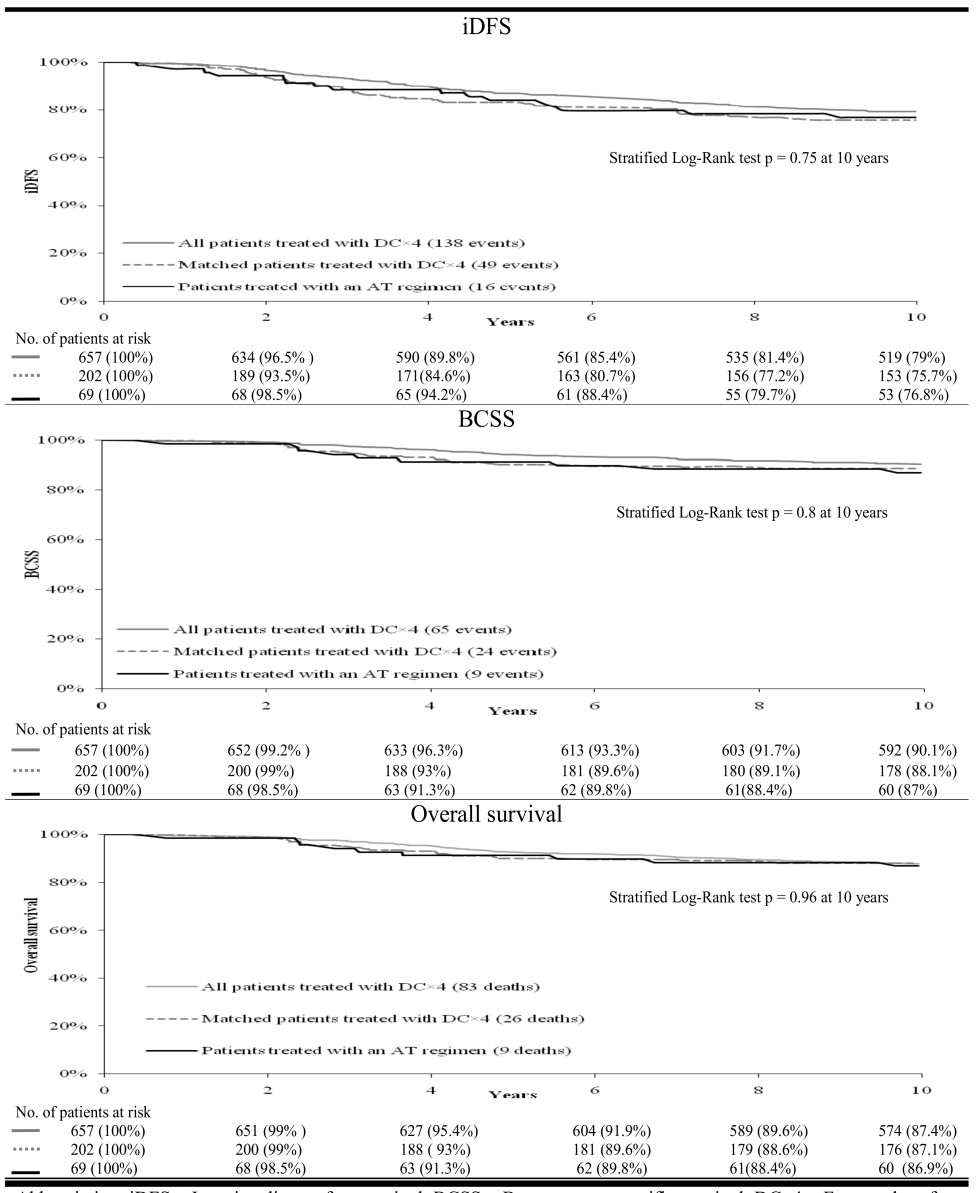

Abbreviation: iDFS = Invasive disease free survival; BCSS = Breast cancer specific-survival; DC×4 = Four cycles of docetaxel-cyclophosphamide; AT= Anthracycline-taxane regimen.

**Figure 1.** Analyses comparing patients treated with DC4 to patients treated with AT.

There was no statistically significant difference for the treatment effects of matched DC4 patients compared to the AT patients on iDFS (−1.1% 10-year difference in iDFS, 75.7% vs. 76.8%, *p* = 0.75, Figure 1; HR = 1.05, 95% CI = 0.65 to 1.8, Table S1), BCSS (+1.1% 10-year difference in BCSS, 88.1% vs. 87%, *p* = 0.8, Figure 1; HR = 0.91, 95% CI = 0.42 to 1.9, Table S1), or OS (+0.2% difference in 10-year OS, 87.1% vs. 86.9%, *p* = 0.96, Figure 1; HR = 0.98, 95% CI = 0.46 to 2.1, Table S1). In exploratory analysis, no interactions between grade (poorly differentiated vs. non-poorly differentiated), stage (I vs. II), or hormone receptor status (positive vs. negative) and type of adjuvant chemotherapy (DC4 vs. AT) were identified.

Unadjusted and adjusted Cox proportional-hazard regression analyses that compared the iDFS, BCSS and OS of all patients treated with DC4 (n = 657) to the case group of 69 patients treated with AT revealed similar results (Figure 1 and Table S1). No interactions between grade, stage, or hormone receptor status and type of adjuvant chemotherapy (DC4 vs. AT) were identified.

## 4. Discussion

To our knowledge, this is the first provincial-based, real world study comparing long term outcomes of patients with HER2-negative, LNN breast cancer treated with DC4 versus an AT regimen. We found that patients treated with AT had more high risk prognostic factors in comparison to the overall population of patients treated with DC; hence, AT patients were matched with similar DC4 patients. The 10-year risk of recurrence seemed to favor AT with a 1.1% absolute difference in iDFS, whereas the 10-year risk of death from breast cancer and any cause seemed to favor DC4 with 1.1% and 0.2% absolute differences in BCSS and OS, respectively. However, statistical significance was not achieved for any of these comparisons.

Prospective randomized clinical trials have compared DC6 with AT regimens for adjuvant treatment of HER2-negative breast cancer and have reported outcomes with shorter follow-up periods [7–9]. The ABC trials included patients with LNN and lymph node positive (LNP) disease. In the pooled analysis of all patients, DC6 was not non-inferior to AT with respect to 4-year iDFS (−2.5%; HR 1.20, 95% CI 0.97–1.49) [7]. Although planned exploratory tests for treatment interaction by nodal status and hormone receptor status were negative, for patients with LNN disease, the overall HR for iDFS was 1.03, 95% CI 0.74–1.44 [7]. The Plan B trial also examined DC6 versus AT in HER2-negative, LNN and LNP breast cancer and found that DC6 was non-inferior to AT in terms of 5-year DFS and OS [8]. Finally, in the Hellenic Oncology Research Group (HORG) trial, only patients with HER2-negative, LNP breast cancer were included. As per the ABC trials, DC6 was not non-inferior to AT with respect to 3-year DFS (−1.6%; HR 1.147, 95% CI 0.716–1.839) [9]. Recently, a pooled analysis of the ABC, Plan B and HORG trials has been reported. Neither the superiority of AT nor the non-inferiority of DC6 was established [10]. The difference in 5-year DFS was small (−1.38%; HR 1.11, 95% CI 0.95–1.30) [10].

Although this is a retrospective cohort study, we used the rigorous linkage of high quality population data from comprehensive health databases and we are reporting with a long median follow-up time of nine years. We have taken care to avoid sources of bias and confounding by conducting a matched cohort analysis. We were able to consider most potential prognostic variables; however, other factors that could have impacted survival outcomes were not included, such as patient menopausal status, tumour lymphovascular invasion, chemotherapy dose reduction and/or delay, and type, duration and adherence to endocrine therapies.

Despite the lack of statistical power in our analysis to rule out small benefits for one regimen versus the other, the similar 10-year iDFS, BCSS and OS rates for the matched DC4 and AT groups are reassuring and provide evidence to support the ongoing use of adjuvant DC4 for HER2-negative, LNN breast cancer. Further analyses of other real world datasets are warranted to validate our exploratory subgroup analysis by hormone receptor status.

**Supplementary Materials:** The following are available online at https://www.mdpi.com/1718-772 9/28/2/109/s1, Table S1: Cox regression analyses for iDFS, BCSS, and OS.

**Author Contributions:** Conceptualization, M.H., A.B. and S.L.; methodology, M.H., A.B. and S.L.; formal analysis, M.H.; data curation, M.H., A.B. and S.L.; writing—original draft preparation, M.H.; writing—review and editing, A.B. and S.L. All authors have read and agreed to the published version of the manuscript.

**Funding:** This research received no external funding.

**Institutional Review Board Statement:** The study was institutionally approved under the Alberta Research Ethics Community Consensus Initiative as minimal risk 29 May 2020.

**Informed Consent Statement:** Patient consent was waived as protocol minimal risk and population-based outcomes were sought.

**Data Availability Statement:** The data presented in this study are available on request from the corresponding author. The data are not publicly available due to privacy.

**Conflicts of Interest:** The authors declare no conflict of interest.

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
