# Peer review of "Four Cycles of Docetaxel-Cyclophosphamide versus Anthracycline-Taxane as Adjuvant Chemotherapy for HER2-Negative, Axillary Lymph Node Negative Breast Cancer: A Real-World Comparison of Alberta Patients Treated 2008–2012"

_curroncol, doi:10.3390/curroncol28020109_

Round 1

Reviewer 1 Report

The authors should report in the Table 1 and analyze patients in pre and post menopausal status

The Authors should report and describe the adjuvant endocrine treatment: type and duration

Author Response

Dear Reviewer,

Many thanks for taking the time to review our manuscript and provide feedback. Please see our responses to your comments.

  1. “The authors should report in the Table 1 and analyze patients in pre and post menopausal status”

We agree with the reviewer that menopausal status is an important potential prognostic variable to be considered. Unfortunately, menopausal status is not captured by the database used for the purpose of this project and was not recorded during the chart review. Therefore, we have highlighted this limitation in lines 155-156 of the discussion section “however, other factors that could have impacted survival outcomes were not included such as patient menopausal status, tumour lymphovascular invasion, chemotherapy dose reduction and/or delay, and adherence to endocrine therapies.”

  1. “The Authors should report and describe the adjuvant endocrine treatment: type and duration”

We also agree with the reviewer that type and duration of endocrine therapy is an important prognostic variable to be considered in our analysis. Unfortunately, obtaining type of hormonal therapy and duration with reliability requires laborious chart review and cross-referencing with the pharmacy database. The Alberta Cancer Registry, which provides information to the Breast Data Mart, does flag patients who have had a hormonal therapy prescription associated with their initial diagnosis. Amongst the hormone receptor-positive AT patients, 75% had a hormonal therapy flag and amongst the hormone receptor-positive DC patients, 84% had a hormonal therapy flag. The accuracy of this flag is unknown and hence, we do not feel comfortable reporting this additional data. We have further revised our sentence in lines 155-156 to clarify further this limitation and it now reads: “however, other factors that could have impacted survival outcomes were not included such as patient menopausal status, tumour lymphovascular invasion, chemotherapy dose reduction and/or delay, and type, duration and adherence to endocrine therapies.”

Reviewer 2 Report

Dear authors, the report is timely and interesting. The findings are consistent with consolidated knowledge in a controversial topic.

The population is small, however 50/50 patients are tnbc/luminal-like. Therefore, it might be relevant to consider a sub-analysis, totally explorative, to understand possible signals of interactions with HR-status. Subanalsis also useful for grade 3 Vs non-grade 3 and pT3 vs non-pT3/ stage I vs stage II. 

Though not easy, a descriptive analysis of the adjuvant hormone therapies can be useful, eg: tamoxifen (n/N), AI (n/N), etc. 

It is unclear if all the patients have received FEC100 or different doses of anthracyclines. While limits in the report of dose density/intensity have been outlined by the authors, a mention on the "intended" regimens would be helpful. 

The cohort is cut at 2012, so no post-adjuvant treatment should have been available at that time - please specify. 

Report the menopausal status in the cohorts, mark possible differences. Also, ovarian function suppression should be mentioned. 

Author Response

Dear Reviewer,

Please see our responses to your comments.

1. “Dear authors, the report is timely and interesting. The findings are consistent with consolidated knowledge in a controversial topic.”

We would like to take this opportunity to thank you for taking the time to read and provide feedback on our work.

2. “The population is small, however 50/50 patients are tnbc/luminal-like. Therefore, it might be relevant to consider a sub-analysis, totally explorative, to understand possible signals of interactions with HR-status. Subanalsis also useful for grade 3 Vs non-grade 3 and pT3 vs non-pT3/ stage I vs stage II”

We conducted exploratory analyses to test the interaction between a given tumour characteristic (grade, stage, and hormone receptor status) and type of adjuvant chemotherapy (DC4 vs. AT) for significance and found none. To clarify these exploratory analyses to readers, we added the following paragraph to the Statistical Analysis section (lines 95-98) “In all above analyses, we also examined whether the association of adjuvant chemotherapy (DC4 vs. AT) with 10-year iDFS, BCSS, and OS is different by tumour characteristics (grade, stage and hormone receptor status) by testing for interactions between these tumour characteristics and the type of adjuvant chemotherapy (DC4 vs. AT).” We also added the following 2 sentences under the Results section (lines 117-120) “In exploratory analysis, no interactions between grade (poorly differentiated vs. non-poorly differentiated), stage (I vs. II), or hormone receptor status (positive vs. negative) and type of adjuvant chemotherapy (DC4 vs. AT) were identified” and “No interactions between grade, stage, or hormone receptor status and type of adjuvant chemotherapy (DC4 vs. AT) were identified” (lines 123-124). Finally, we edited our last sentence in the Discussion section and it now reads “Further analyses of other real world datasets are warranted to validate our exploratory subgroup analyses by hormone receptor status” (lines 160-161).

3. “Though not easy, a descriptive analysis of the adjuvant hormone therapies can be useful, eg: tamoxifen (n/N), AI (n/N), etc.” 

We agree that endocrine therapy is an important prognostic variable to be considered in our analysis. Unfortunately, obtaining type of hormonal therapy and duration with reliability requires laborious chart review and cross-referencing with the pharmacy database. The Alberta Cancer Registry, which provides information to the Breast Data Mart, does flag patients who have had a hormonal therapy prescription associated with their initial diagnosis. Amongst the hormone receptor-positive AT patients, 75% had a hormonal therapy flag and amongst the hormone receptor-positive DC patients, 84% had a hormonal therapy flag. The accuracy of this flag is unknown and hence, we do not feel comfortable reporting this additional data. We have further revised our sentence in lines 155-156 to clarify further this limitation and it now reads: “however, other factors that could have impacted survival outcomes were not included such as patient menopausal status, tumour lymphovascular invasion, chemotherapy dose reduction and/or delay, and type, duration and adherence to endocrine therapies.”

4. “It is unclear if all the patients have received FEC100 or different doses of anthracyclines. While limits in the report of dose density/intensity have been outlined by the authors, a mention on the "intended" regimens would be helpful.”

Yes, there was some variability in total anthracycline dose although the vast majority of patients received FEC-D with an intended dose of epirubicin of 100 mg/m2 for three doses. We have edited the results (line 103) to read “The most commonly prescribed AT regimen (91.3%) was three cycles of fluorouracil, epirubicin (100 mg/m2), cyclophosphamide followed by three cycles of docetaxel (FEC-D).” The remaining patients were split evenly between AC-D (doxorubicin at 60 mg/m2 for four doses) and DAC (doxorubicin at 50 mg/m2 for 6 doses).

5. The cohort is cut at 2012, so no post-adjuvant treatment should have been available at that time - please specify. 

We are not clear as to which post-adjuvant treatments are being questioned. Adjuvant bisphosphonates would not have been standard of care for postmenopausal women in this era. Please let us know if we can elaborate further.

6. Report the menopausal status in the cohorts, mark possible differences. Also, ovarian function suppression should be mentioned.

We agree that menopausal status is an important potential prognostic variable to be considered. Unfortunately, menopausal status and information on hormonal therapy have not been obtained for this cohort. Therefore, we have highlighted this limitation in lines 155-156 of the discussion section “however, other factors that could have impacted survival outcomes were not included such as patient menopausal status, tumour lymphovascular invasion, chemotherapy dose reduction and/or delay, and adherence to endocrine therapies.”